# Neural Characteristic Activation Value Analysis for Improved ReLU Network Feature Learning

## Abstract

This work examines the characteristic activation values of individual ReLU units in neural networks. We refer to the set of input locations corresponding to such characteristic activation values as the *characteristic activation set* of a ReLU unit. We draw an explicit connection between the characteristic activation set and learned features in ReLU networks. This connection leads to new insights into how various neural network normalization techniques used in modern deep learning architectures regularize and stabilize stochastic gradient optimization. Utilizing these insights, we propose geometric parameterization for ReLU networks to improve feature learning, which decouples the radial and angular parameters in the hyperspherical coordinate system. We empirically verify its usefulness with less carefully chosen initialization schemes and larger learning rates. We report significant improvements in optimization stability, convergence speed, and generalization performance for various models on a variety of datasets, including the ResNet-50 network on ImageNet.

## 1 Introduction

In a neural network with standard parameterization (SP), each neuron applies an affine transformation to its input $\mathbf{x} \in \mathbb{R}^n$ followed by an element-wise nonlinear activation function $g$:

$$z = g(\mathbf{w}^\mathsf{T} \mathbf{x} + b), \tag{1}$$

where the affine transformation is parameterized by a weight vector $\mathbf{w} \in \mathbb{R}^n$ and a bias scalar $b \in \mathbb{R}$. Rectified Linear Unit (ReLU) (Glorot et al., 2011) is arguably the most popular activation function used in modern deep learning architectures, which has a cut-off point at $s = 0$:

$$g(s) = \mathrm{ReLU}(s) = \max(0, s). \tag{2}$$

The characteristic activation boundary/set of such a ReLU neuron refers to the set of input locations with zero pre-activations, which, by definition, separates the active region from the inactive region in the input space. Characteristic activation boundaries are the building blocks for the decision boundaries of ReLU networks, which characterize the quality of the learned features.

Based on the proposed characteristic activation analysis, this paper focuses on a geometric interpretation of learned features in ReLU networks. This provides a theoretical justification for how various neural network normalization techniques used in modern deep learning architectures regularize and stabilize stochastic gradient optimization. Motivated by these insights, we propose a novel neural network parameterization technique that decouples the radial and angular parameters in the hyperspherical coordinate system and thus smooths the evolution of the characteristic activation boundaries in ReLU networks. We empirically show that our new parameterization enables faster and more stable stochastic gradient optimization and achieves better generalization performance even under less carefully chosen initialization schemes and larger learning rates.

## 2 Background and Related Work

This section reviews neural network reparameterization and normalization techniques, paying particular attention to weight normalization and batch normalization.

**Weight normalization (WN)** (Salimans & Kingma, 2016) is a simple weight reparameterization technique that decouples the length $l$ and the direction $\mathbf{v}/\|\mathbf{v}\|_2$ of $\mathbf{w}$ in a standard ReLU unit (1):

$$z = \text{ReLU}\left( l \left( \frac{\mathbf{v}}{\|\mathbf{v}\|_2} \right)^{\text{T}} \mathbf{x} + b \right). \tag{3}$$

The idea behind WN is to make the length $l$ and the direction $\mathbf{v}/\|\mathbf{v}\|_2$ of the weight vector independent of each other in the Cartesian coordinate system, which is effective in improving the conditioning of the gradients of the parameters and speeding up the convergence of optimization.

**Batch normalization (BN)** (Ioffe & Szegedy, 2015) is a widely-used neural network normalization layer in modern deep learning architectures such as ResNet (He et al., 2016), which is effective to accelerate and stabilize stochastic gradient optimization of neural networks (Kohler et al., 2019). In ReLU networks, BN is often applied at the pre-activation level in each layer:

$$z = \text{ReLU}(\text{BN}(\mathbf{w}^{\text{T}} \mathbf{x} + b)). \tag{4}$$

The BN layer standardizes the pre-activation using the empirical mean and covariance estimated from the current mini-batch:

$$\text{BN}(\mathbf{w}^{\text{T}} \mathbf{x} + b) = \gamma \frac{\mathbf{w}^{\text{T}} \mathbf{x} - \hat{\mathbb{E}}_{\mathbf{x}}[\mathbf{w}^{\text{T}} \mathbf{x}]}{\sqrt{\hat{\text{Var}}_{\mathbf{x}}[\mathbf{w}^{\text{T}} \mathbf{x} + b]}} + \beta, \tag{5}$$

where $\gamma \in \mathbb{R}$ and $\beta \in \mathbb{R}$ are two free parameters to be learned from data, which adjusts the output of the BN layer as needed to increase its expressiveness.

**Connections between BN and WN.** BN and WN are closely related to one another: assuming that the input $\mathbf{x}$ has zero means, one can show that BN is also a kind of neural network parameterization:

$$\text{BN}(\mathbf{w}^{\text{T}} \mathbf{x} + b) = \gamma \frac{\mathbf{w}^{\text{T}} \mathbf{x}}{\sqrt{\hat{\text{Var}}_{\mathbf{x}}[\mathbf{w}^{\text{T}} \mathbf{x} + b]}} + \beta = \gamma \frac{\mathbf{w}^{\text{T}} \mathbf{x}}{\sqrt{\mathbf{w}^{\text{T}} \hat{\Sigma} \mathbf{w}}} + \beta = \gamma \left( \frac{\mathbf{w}}{\|\mathbf{w}\|_{\hat{\Sigma}}} \right)^{\text{T}} \mathbf{x} + \beta, \tag{6}$$

where the vector norm $\|\mathbf{w}\|_{\hat{\Sigma}}$ is calculated with respect to the empirical data covariance matrix $\hat{\Sigma} = \hat{\text{Var}}[\mathbf{x}]$ estimated from the current mini-batch. This shows that BN is effectively an adaptive, data-dependent parameterization of standard neurons (1) that decouples the correlations in the input $\mathbf{x}$. However, in practice, it is common to estimate only the diagonal elements in the data covariance matrix $\hat{\Sigma}$ and set all its off-diagonal elements to zero to reduce the extra computation introduced by the BN layers. Under this formalism, WN can be seen as a special case of BN where the covariance matrix $\hat{\Sigma}$ is replaced by the identity matrix $\mathbf{I}$ independent of the input $\mathbf{x}$, since $\|\cdot\|_2 = \|\cdot\|_{\mathbf{I}}$.

**Other normalization methods.** Instead of normalizing the batch dimension as in BN, LayerNorm (Ba et al., 2016) normalizes the feature dimension, which is preferred for small batches of high-dimensional inputs. Other variants of BN include SwitchNorm (Luo et al., 2018) and IEBN (Liang et al., 2020). There are other normalization techniques designed for specific applications, e.g., instance normalization (Ulyanov et al., 2016) and group normalization (Wu & He, 2018) are special cases of instance normalization designed for CNNs, and spectral normalization (Miyato et al., 2018; Zhai et al., 2023) is specifically designed for GANs and transformers.

## 3 CHARACTERISTIC ACTIVATION VALUE ANALYSIS FOR ReLU NETWORKS

This section formally defines the characteristic activation sets of individual neurons and introduces a geometric connection between such sets and learned features in ReLU networks. This geometric insight will help understand the stability of neural network optimization and motivate a new neural network parameterization that is provably stable under stochastic gradient optimization.

### 3.1 CHARACTERISTIC ACTIVATION SETS FOR ReLU UNITS

**Definition 3.1.** The ReLU activation function (2) is active for positive arguments $s > 0$ and inactive for negative arguments $s < 0$. For a neuron with ReLU activation, the *characteristic activation set* $\mathcal{B}$ is defined by a set of input locations such that $s = 0$:

$$\mathcal{B} = \{\mathbf{x} \in \mathbb{R}^n : \mathbf{w}^{\text{T}} \mathbf{x} + b = 0\}. \tag{7}$$

In other words, it forms a *characteristic boundary* $\mathcal{B}$ for each neuron, which is an $(n-1)$-dimensional hyperplane that separates the active and inactive regions of a ReLU unit in the input space $\mathbb{R}^n$.

**Definition 3.2.** We define a representative point $\phi$ that lies on the characteristic boundary $\mathcal{B}$ as

$$\phi = -\frac{b\,\mathbf{w}}{\mathbf{w}^{\mathrm{T}}\,\mathbf{w}} = -\frac{b}{\|\mathbf{w}\|_2}\frac{\mathbf{w}}{\|\mathbf{w}\|_2}. \tag{8}$$

We refer to the point $\phi$ as the *spatial location* of $\mathcal{B}$ and the vector that goes from the origin to the point $\phi$ as the *characteristic vector* of $\mathcal{B}$ (i.e., shortest path between the origin and $\mathcal{B}$). The spatial location (or the characteristic vector) $\phi$ uniquely determines the characteristic set/boundary.

## 3.2 ReLU Characteristic Activation Boundary in Hyperspherical Coordinate

In a high dimensional input space, most data points $\mathbf{x}$ live in a thin shell since the volume of a high dimensional space concentrates near its surface (Blum et al., 2020). Intuitively, we want the spatial locations $\phi$ of characteristic activation boundaries $\mathcal{B}$ to be close to the thin shell where most data points lie, because this spatial affinity between the characteristic activation set and data points will introduce non-linearity at suitable locations in the input space to separate different inputs $\mathbf{x}$ by assigning them different activation values. This motivates the use of the hyperspherical coordinate to represent the spatial locations of the characteristic activation boundaries.

More concretely, we reparameterize the characteristic activation boundary in terms of its characteristic radius $\lambda \in \mathbb{R}$ and angle $\boldsymbol{\theta} = [\theta_1, \cdots, \theta_{n-1}]^{\mathrm{T}}$ in the hyperspherical coordinate system. Noticing that $\mathbf{w}/\|\mathbf{w}\|_2$ is a unit vector, the radial-angular decomposition of the characteristic vector is given by

$$\phi(\lambda, \boldsymbol{\theta}) = -\lambda\,\mathbf{u}(\boldsymbol{\theta}), \quad \text{with the definition } \lambda := \frac{b}{\|\mathbf{w}\|_2} \text{ and } \mathbf{u}(\boldsymbol{\theta}) := \frac{\mathbf{w}}{\|\mathbf{w}\|_2}, \tag{9}$$

where the direction of the unit vector $\mathbf{u}(\boldsymbol{\theta})$ is determined by the characteristic angle $\boldsymbol{\theta}$:

$$\mathbf{u}(\boldsymbol{\theta}) = \begin{bmatrix} \cos(\theta_1) \\ \sin(\theta_1)\cos(\theta_2) \\ \sin(\theta_1)\sin(\theta_2)\cos(\theta_3) \\ \vdots \\ \sin(\theta_1)\sin(\theta_2)\sin(\theta_3)\cdots\sin(\theta_{n-2})\cos(\theta_{n-1}) \\ \sin(\theta_1)\sin(\theta_2)\sin(\theta_3)\cdots\sin(\theta_{n-2})\sin(\theta_{n-1}) \end{bmatrix} \in S^{n-1}, \tag{10}$$

where $S^{n-1} := \{\mathbf{x} \in \mathbb{R}^n : \|\mathbf{x}\|_2 = 1\}$ is the unit hypersphere in $\mathbb{R}^n$. In the hyperspherical coordinate system, the characteristic activation boundary can be expressed as

$$\mathcal{B}(\lambda, \boldsymbol{\theta}) = \{\mathbf{x} \in \mathbb{R}^n : \mathbf{u}(\boldsymbol{\theta})^{\mathrm{T}}\mathbf{x} + \lambda = 0\}. \tag{11}$$

## 3.3 Geometric Interpretation of ReLU Characteristic Activation Set

The characteristic activation set $\mathcal{B}$ of a ReLU Unit forms a line in $\mathbb{R}^2$, as shown by the brown solid line in Figure 1a. More generally, $\mathcal{B}$ forms an $(n-1)$-dimensional hyperplane in $\mathbb{R}^n$. The spatial location/characteristic vector $\phi = -\lambda\,\mathbf{u}(\boldsymbol{\theta})$ fully specifies the characteristic activation boundary $\mathcal{B}$: it is perpendicular to $\mathcal{B}$, and its endpoint lies on $\mathcal{B}$. The angle $\boldsymbol{\theta}$ controls the direction of the characteristic activation boundary. The radius $\lambda$ controls the distance between the origin and the characteristic activation boundary. Geometrically speaking, calculating the pre-activation of a ReLU unit for an input $\mathbf{x}$ is equivalent to projecting $\mathbf{x}$ onto the unit vector $\mathbf{u}(\boldsymbol{\theta})$ and then adding the radius $\lambda$ to the signed norm of the projected vector. From this perspective, it is clear the characteristic activation boundary is a set of inputs whose projections over $\mathbf{u}(\boldsymbol{\theta})$ have signed norm $-\lambda$. For this reason, we call this radial-angular decomposition in the hyperspherical coordinate system the *geometric parameterization* (GmP).

## 3.4 Perturbation Analysis of ReLU Characteristic Activation Boundary

One benefit of defining characteristic activation boundaries in the hyperspherical coordinate system is that the radius $\lambda$ and angle $\boldsymbol{\theta}$ of the spatial location $\phi$ are disentangled. More concretely, this means that small perturbations to the parameter $\lambda$ and $\boldsymbol{\theta}$ will only cause small changes in the spatial location of the characteristic activation boundary. To illustrate this, first, we consider a small perturbation $\boldsymbol{\varepsilon}$ (e.g., gradient noise during SGD) to the weight $\mathbf{w}$ in the standard parameterization (SP). This perturbation results in a change in the angular direction of the characteristic activation boundary by

$$\langle \mathbf{w}, \mathbf{w} + \boldsymbol{\varepsilon} \rangle = \arccos\left(\frac{\mathbf{w}^{\mathrm{T}}(\mathbf{w} + \boldsymbol{\varepsilon})}{\|\mathbf{w}\|_2\|\mathbf{w} + \boldsymbol{\varepsilon}\|_2}\right), \tag{12}$$

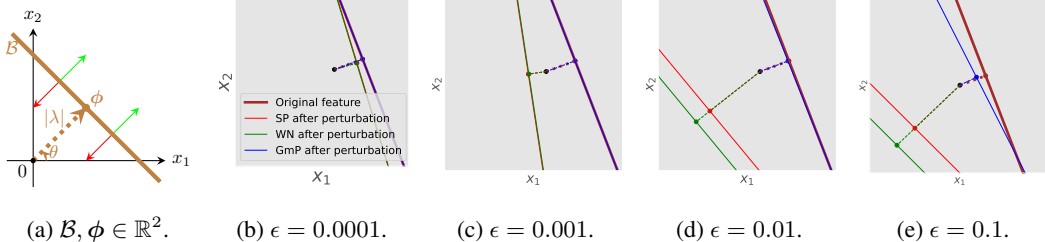

(a) $\mathcal{B}, \phi \in \mathbb{R}^2$.  (b) $\epsilon = 0.0001$.  (c) $\epsilon = 0.001$.  (d) $\epsilon = 0.01$.  (e) $\epsilon = 0.1$.

Figure 1: (a) Characteristic activation boundary $\mathcal{B}$ (brown solid line) and spatial location $\phi = -\lambda \mathbf{u}(\theta)$ of a ReLU unit $z = \text{ReLU}(\mathbf{u}(\theta)^{\text{T}} \mathbf{x} + \lambda) = \text{ReLU}(\cos(\theta)x_1 + \sin(\theta)x_2 + \lambda)$ for inputs $\mathbf{x} \in \mathbb{R}^2$. The characteristic activation set forms a line in $\mathbb{R}^2$, which acts as a boundary separating inputs into two regions. Green arrows denote the active region, and red arrows denote the inactive region. (b)-(e) Stability of the characteristic activation boundary (set) of a ReLU unit in $\mathbb{R}^2$ under small perturbations $\boldsymbol{\varepsilon} = \epsilon \mathbf{1}$ to the parameters of the ReLU unit. Solid lines denote characteristic activation boundaries $\mathcal{B}$, and colored dotted lines connect the origin and spatial locations $\phi$ of $\mathcal{B}$. Smaller changes between the perturbed and original boundaries imply higher stability. GmP is most stable against perturbations.

which can take arbitrary values in $[0, \pi]$ even for a small perturbation $\boldsymbol{\varepsilon}$. For example, we could have $\langle \mathbf{w}, \mathbf{w} + \boldsymbol{\varepsilon} \rangle = \pi$ for $\boldsymbol{\varepsilon} = -(1 + \epsilon) \mathbf{w}$, $\forall \epsilon > 0$. This indicates that the characteristic activation boundary is unstable in the sense that it is vulnerable to small perturbations if the weight $\mathbf{w}$ has a small norm, which is the case during neural network training since large weights would lead to overfitting and numerical instability (e.g., the widely-used weight decay method explicitly regularizes $\|\mathbf{w}\|_2$ to be close to zero). This has the implication that even a small gradient noise could destabilize the evolution of characteristic boundaries during stochastic gradient optimization. Such instability is a critical reason that prevents practitioners from using larger learning rates (Goodfellow et al., 2016).

In contrast, our GmP in the hyperspherical coordinate system is much more stable under perturbation: we show that the change in the angular direction $\langle \mathbf{u}(\boldsymbol{\theta}), \mathbf{u}(\boldsymbol{\theta} + \boldsymbol{\varepsilon}) \rangle$ of the characteristic activation boundary $\mathcal{B}$ under perturbation $\boldsymbol{\varepsilon}$ is bounded by the magnitude of perturbation $\boldsymbol{\varepsilon}$.

**Theorem 3.3.** *With a small perturbation $\boldsymbol{\varepsilon} := [\varepsilon_1, \cdots, \varepsilon_{n-1}]^T$ to the angular parameter $\boldsymbol{\theta}$, the change in the angular direction $\mathbf{u}(\boldsymbol{\theta}) \in S^{n-1}$ $(n \geq 2)$ of the weights under GmP is given by*

$$\langle \mathbf{u}(\boldsymbol{\theta}), \mathbf{u}(\boldsymbol{\theta} + \boldsymbol{\varepsilon}) \rangle = \sqrt{\varepsilon_1^2 + \sum_{i=2}^{n-1} \left( \prod_{j=1}^{i-1} \sin^2(\theta_j) \right) \varepsilon_i^2} \leq \|\boldsymbol{\varepsilon}\|_2. \tag{13}$$

The proof of Theorem 3.3 can be found in Appendix B, which is based on an elegant idea from differential geometry that the change in the angular direction is simply the norm of the perturbation with respect to the metric tensor $\mathbf{M}$ for the hyperspherical coordinate: $\langle \mathbf{u}(\boldsymbol{\theta}), \mathbf{u}(\boldsymbol{\theta} + \boldsymbol{\varepsilon}) \rangle = \|\boldsymbol{\varepsilon}\|_{\mathbf{M}}$. Under GmP, this metric tensor turns out to be diagonal: $\mathbf{M} = \text{diag}(1, m_{2,2}, \cdots, m_{n-1,n-1})$ with $m_{i,i} = \prod_{j=1}^{i-1} \sin^2(\theta_j) \in [0, 1]$, and thus $\langle \mathbf{u}(\boldsymbol{\theta}), \mathbf{u}(\boldsymbol{\theta} + \boldsymbol{\varepsilon}) \rangle \leq \|\boldsymbol{\varepsilon}\|_2$. This shows that GmP essentially acts as a pre-conditioner, making neural network optimization robust against small perturbations. It might be tempting to think that GmP is identical to WN. Indeed, GmP inherits the advantages of WN because the length-directional decomposition in WN is automatically inherent in GmP. However, GmP possesses an extra nice property that WN lacks: directly parameterizing the angle $\boldsymbol{\theta}$ in the hyperspherical coordinate makes the evolution of the characteristic activation boundaries smoother and more robust against small perturbations (e.g., SGD noise) to the parameters regardless of how small $\|\mathbf{w}\|_2$ is, as shown in Equation (13). In contrast, the characteristic activation boundary under WN is as unstable as SP, since its change in direction under perturbations $(\boldsymbol{\varepsilon}, \varepsilon')$ to $(\mathbf{v}, l)$ is given by

$$\left\langle l \frac{\mathbf{v}}{\|\mathbf{v}\|_2}, (l + \varepsilon') \frac{\mathbf{v} + \boldsymbol{\varepsilon}}{\|\mathbf{v} + \boldsymbol{\varepsilon}\|_2} \right\rangle = \arccos \left( \frac{\mathbf{v}^{\text{T}}(\mathbf{v} + \boldsymbol{\varepsilon})}{\|\mathbf{v}\|_2 \|\mathbf{v} + \boldsymbol{\varepsilon}\|_2} \right), \tag{14}$$

which has exactly the same form as that for SP as in Equation (12). Furthermore, this implies that any weight-space parameterization and normalization techniques will suffer from this issue.

### 3.5 VERIFICATION OF THE HYPOTHESES OF CHARACTERISTIC ACTIVATION ANALYSIS

This section verifies the validity of the hypotheses of our proposed characteristic activation analysis on three illustrative experiments aided with visualization, and demonstrates that the improved stability under GmP is beneficial for neural network optimization and generalization.

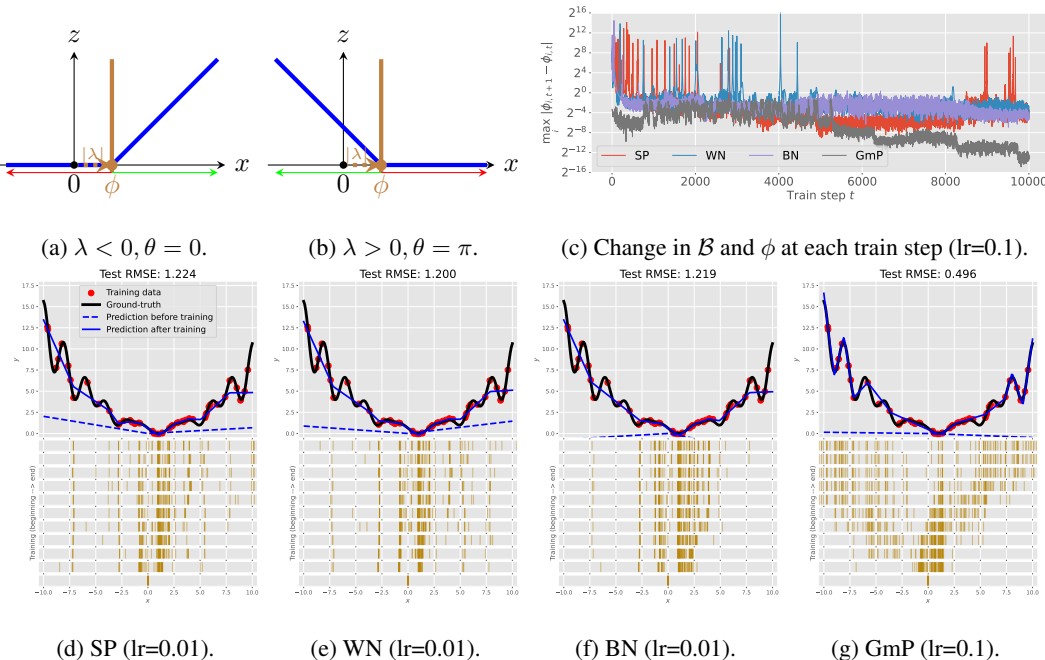

(a) $\lambda < 0, \theta = 0$.     (b) $\lambda > 0, \theta = \pi$.     (c) Change in $\mathcal{B}$ and $\phi$ at each train step (lr=0.1).

(d) SP (lr=0.01).     (e) WN (lr=0.01).     (f) BN (lr=0.01).     (g) GmP (lr=0.1).

Figure 2: (a)-(b) Characteristic activation point $\mathcal{B}$ (intersection of brown solid lines and the x-axis) and spatial location $\phi = -\lambda u(\theta)$ of a single ReLU unit $z = \mathrm{ReLU}(u(\theta)x + \lambda)$ (blue solid lines) for inputs $x \in \mathbb{R}$. Green arrows denote active regions, and red arrows denote inactive regions. (c) Evolution dynamics of the characteristic points $\mathcal{B}$ in a one-hidden-layer network with 100 ReLU units for a 1D Levy regression problem under 4 different parameterizations during training. Smaller values are better as they indicate higher stability of the evolution of the characteristic points during training. The y-axis is in $\log_2$ scale. (d)-(g): The top row illustrates the experimental setup, including the network's predictions at initialization and after training, as well as the training data and the ground-truth function (Levy). A single-hidden-layer network with 100 ReLU units is trained using Adam. Bottom row: the evolution of the characteristic activation point for the 100 ReLU units during training. Each horizontal bar shows the spatial location spectrum for a chosen optimization step, moving from the bottom (at initialization) to the top (after training with Adam). More spread of the spatial locations covers the data better and adds more useful non-linearity to the model, making prediction more accurate. Regression accuracy is measured by root mean squared error (RMSE) on a separate test set. Smaller RMSE values are better. We use cross-validation to select the learning rate for each method. It turns out that the optimal learning rate for SP, WN, and BN is lower than that for GmP, since their training becomes unstable with higher learning rates, as shown in (c).

In Figures 1b-1e, we simulate the evolution behavior of characteristic boundaries in $\mathbb{R}^2$ for three different neural network parameterizations: SP, WN and GmP. We apply small perturbations $\varepsilon$ of different scales $\epsilon$ to the network parameters under different parameterizations and show how it affects the spatial location of the characteristic activation plane. We can see that the characteristic activation plane changes smoothly under GmP as it gradually moves away from its original spatial location as we increase $\epsilon$. In sharp contrast, even a small perturbation $\epsilon$ of magnitude $10^{-3}$ can drastically change the spatial locations of the characteristic activation planes under other parameterizations.

In Figure 2, we train a one-hidden-layer network with 100 ReLU units under various parameterizations on the 1D Levy regression dataset using Adam (Kingma & Ba, 2014). As shown in Figures 2a-2b, $\mathcal{B}$ and $\phi$ reduce to the same point in $\mathbb{R}$, which will be referred to as the characteristic activation point. The angle $\theta$ of the characteristic activation point can only take two values 0 or $\pi$, corresponding to the two directions on the real line. Clearly, GmP significantly improves the stability of the evolution of the characteristic activation point and allows us to use a $10\times$ large learning rate. Figure 2c shows that under GmP the maximum change $\max_i |\Delta\phi_{i,t}| = \max_i |\phi_{i,t+1} - \phi_{i,t}|$ at each train step $t$ is always smaller than 1 throughout training, while under other parameterizations the changes can be up to $2^{16}$ at some steps. The stable evolution of the characteristic point under GmP leads to improved generalization performance on this regression task, as shown in Figures 2d-2g.

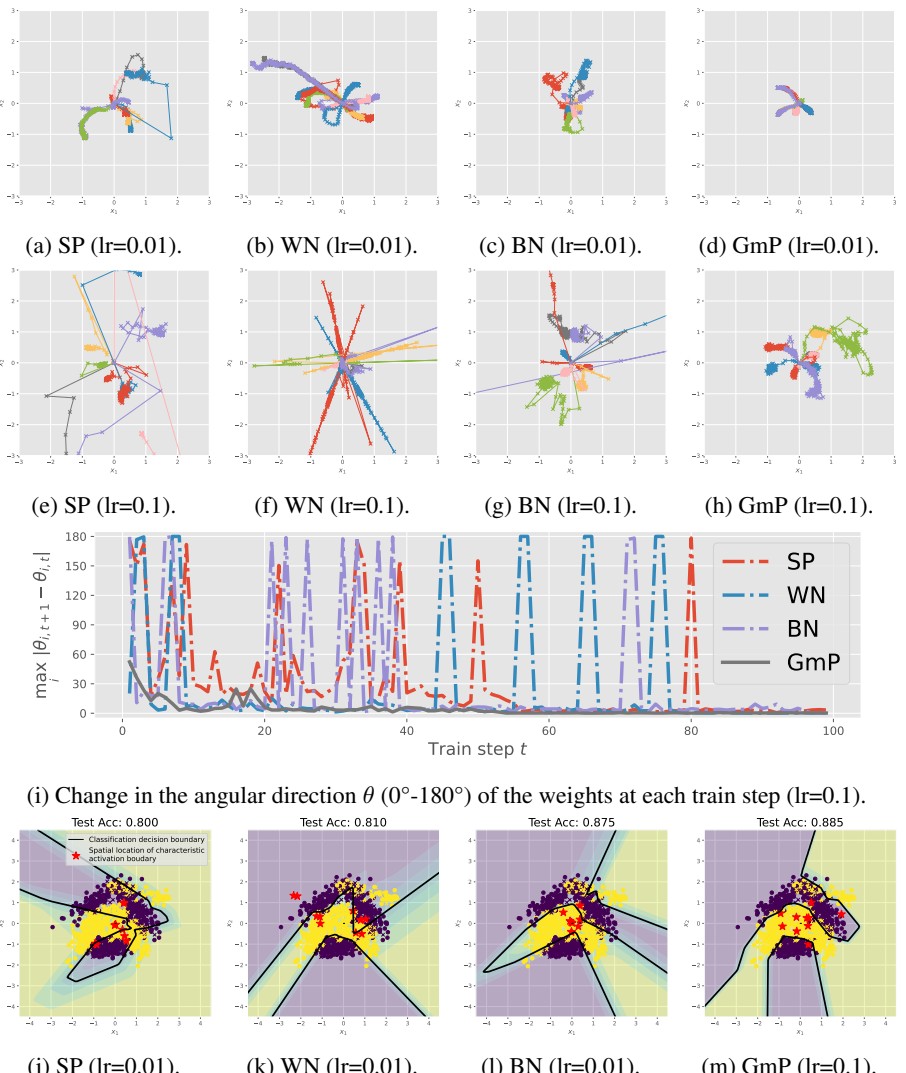

Figure 3: Performance of a single-hidden-layer neural network with 10 ReLU units on the 2D Banana classification dataset under four different parameterizations trained using Adam. (a)-(h): Trajectories of the spatial locations of the 10 ReLU units during training. Each color depicts one ReLU unit. Smoother evolution means higher training stability. The evolution under GmP is stable, so we can use a $10\times$ larger learning rate. (i): Evolution dynamics of the angles $\theta$ of the weights. Smaller values are better as they indicate higher robustness against stochastic gradient noise. (j)-(m): Network predictions after training. Black bold lines depict the classification boundary between two classes. Classification accuracy is measured on a separate test set. Higher accuracy values are better. The red stars show the spatial locations of 10 ReLU units. Intuitively speaking, more evenly spread out red stars are better for classification accuracy, as they provide more useful non-linearity.

In Figure 3, we train a one-hidden-layer network with 100 ReLU units under various parameterizations on the 2D Banana classification dataset using Adam. Figures 3a-3h show that GmP allows us to use a $10\times$ larger learning rate while maintaining a smooth evolution of the characteristic activation boundary. Figure 3i shows that GmP is the only method that guarantees stable updates for the angular directions of the weights during training with a large learning rate: under GmP, the maximum change $\max_i |\Delta\theta_{i,t}| = \max_i |\theta_{i,t+1} - \theta_{i,t}|$ at each train step $t$ remains low throughout training, while under other parameterizations the change can be up to $180°$ at some steps. This verifies the hypothesis in our proposed perturbation analysis. Figures 3j-3m show that under GmP, the spatial locations of the characteristic activation boundaries move towards different directions during training and spread over all training data points in different regions, which forms a classification decision boundary with a reasonable shape that achieves the highest test accuracy among all compared methods.

## 4 GEOMETRIC PARAMETERIZATION FOR RELU NETWORKS

Motivated by the characteristic activation analysis in the hyperspherical coordinate system, this section formally presents geometric parameterization (GmP) for ReLU networks.

### 4.1 GEOMETRIC PARAMETERIZATION FOR RELU UNITS

Starting from reparameterizing a single ReLU unit, we replace the weight vector $\mathbf{w} \in \mathbb{R}^n$ and the bias scalar $b \in \mathbb{R}$ in a standard ReLU unit (1) using the radial parameter $\lambda \in \mathbb{R}$ and the angular vector $\boldsymbol{\theta}$ as defined in Equations (9) and (10). We denote the activation scale by $r$ and move it to the outside of the ReLU activation function. These changes lead to the geometric parameterization (GmP), a new general-purpose parameterization for ReLU networks:

$$z = r \operatorname{ReLU}(\mathbf{u}(\boldsymbol{\theta})^{\mathrm{T}} \mathbf{x} + \lambda). \tag{15}$$

GmP has three learnable parameters: the scaling parameter $r$, the radial parameter $\lambda$, and angular parameter $\boldsymbol{\theta} \in [\theta_1, \cdots, \theta_{n-1}]$ (i.e., $n + 1$ degrees of freedom in total, which is the same as SP). As discussed in Section 3.3, $\lambda$ and $\boldsymbol{\theta}$ specify the spatial location $\phi$ of the characteristic activation boundary. The scaling parameter $r$ determines the scale of the activation. As we have seen in Section 3, GmP results in several nice properties to feature learning: optimizing these geometric parameters in the hyperspherical coordinate during training directly translates into a smooth evolution of the spatial location of the characteristic activation boundary and the scale of the activation.

Let $n$ and $m$ denote the fan-in and fan-out of a layer. Compared to SP, GmP needs to additionally compute $2n - 2$ scalars $\sin(\theta_1), \cdots, \sin(\theta_{n-1}), \cos(\theta_1), \cdots, \cos(\theta_{n-1})$ for each of the $m$ neurons. The cost of these computations is $\mathcal{O}(mn)$ for all neurons in each layer. However, since the cost of computing the affine transformation for each layer is also $\mathcal{O}(mn)$, the total computational cost of GmP remains $\mathcal{O}(mn)$ for each layer, which is the same as SP.

We apply GmP to all layers except for the output layer. The output layer is a linear layer with an inverse link function (e.g., softmax or identity) for producing the network output. Since the inverse link function involves no feature learning, the output layer cannot be reparameterized. For multiple-hidden-layer networks, the inputs to immediate layers are outputs from previous layers, potentially suffering from a covariate shift phenomenon (Salimans & Kingma, 2016). The next section presents a simple fix by normalizing the input means to ReLU units under GmP.

### 4.2 INPUT MEAN NORMALIZATION FOR INTERMEDIATE LAYERS

One implicit assumption of the characteristic activation set analysis is that the input distribution to a neuron centers around the origin during training. This assumption automatically holds for one-hidden-layer networks since the training data is constant. However, this assumption is not necessarily satisfied for the inputs to the intermediate layers in a multiple-hidden-layer network. This is because the inputs to an intermediate layer are transformed by the weights and squashed by the activation function in the previous layer, which could cause optimization difficulties even under GmP due to covariant shift. For ReLU units in the intermediate layers of a multiple-hidden-layer, we propose a simple fix called input mean normalization (IMN), which subtracts the input by their empirical mean:

$$z = r \operatorname{ReLU}(\mathbf{u}(\boldsymbol{\theta})^{\mathrm{T}}(\mathbf{x} - \hat{\mathbb{E}}[\mathbf{x}]) + \lambda). \tag{16}$$

This is a parameter-free data pre-processing technique which centers the inputs around the origin. Although the mean-only batch normalization (MBN) (Salimans & Kingma, 2016) for WN is similar to our IMN, MBN cannot address the covariate shift problem in GmP as it is applied to pre-activations.

### 4.3 LAYER-SIZE INDEPENDENT PARAMETER INITIALIZATION

While existing neural network parameterizations are sensitive to initialization, GmP can work with less carefully chosen initialization schemes independent of the width of the layer, thanks to an invariant property of the hyperspherical coordinate system. To see this, first, we consider the distribution of the angular direction of the characteristic activation boundary under SP. Under popular initialization methods such as the Glorot initialization (Glorot & Bengio, 2010) and He initialization (He et al., 2015), each element in the initial weight vector $\mathbf{w}$ in the SP is independently and identically sampled from a zero mean Gaussian distribution with a layer-size dependent variance. However, this always

induces a uniform distribution over the unit $n$-sphere for the direction $\mathbf{u}(\boldsymbol{\theta})$ of the characteristic activation boundary, no matter what variance value is used in that Gaussian distribution. This allows us to initialize the angular parameter $\boldsymbol{\theta}$ uniformly at random. The parameter $\lambda$ is initialized to zero due to its connection $\lambda = {}^b/{\|w\|_2}$ to SP and the common practice to set $b = 0$ at initialization. The scaling parameter $r$ is initialized to one based on the intuition that the scale $r$ roughly corresponds to the total variance of the weights $\mathbf{w}$ in SP. Therefore, none of the parameters $\lambda$, $\boldsymbol{\theta}$, and $r$ in GmP require layer-size dependent initialization.

## 5 EXPERIMENTS

Section 3.5 already presented a detailed analysis of GmP aided with visualization on three illustrative experiments and clearly demonstrated its improved stability and generalization performance. This section further evaluates the performance of GmP on more challenging real-world machine learning benchmarks, including ImageNet. We apply GmP to several popular deep learning architectures, including ResNet-50, and train them with various widely-used optimizers, including SGD and Adam. We use cross-validation to select the best learning rate for each compared method in every experiment. A more detailed setup for each experiment can be found in Appendix C.

**ImageNet classification with ResNet-50.** We evaluate GmP with a gold-standard large residual neural network ResNet-50 (He et al., 2016) on the ImageNet (ILSVRC 2012) dataset (Deng et al., 2009), which consists of 1,281,167 training images and 50,000 validation images that contain objects from 1,000 categories. The size of the images ranges from $75 \times 56$ to $4288 \times 2848$. We follow exactly the same experimental setup for optimization and data augmentation

Table 1: Validation accuracy (%) for ResNet-50 trained on ImageNet.

| Metric | Top-1 valid. acc. | Top-5 valid. acc. |
|---|---|---|
| WN+MBN | $72.58 \pm 0.16$ | $90.77 \pm 0.12$ |
| BN | $73.63 \pm 0.09$ | $91.11 \pm 0.05$ |
| **GmP+IMN** | $\mathbf{75.57 \pm 0.12}$ | $\mathbf{92.68 \pm 0.11}$ |

as in He et al. (2016). Specifically, we use the SGD optimizer with momentum 0.9, which turns out to be better than Adam for image classification tasks (He et al., 2016). We reduce the learning rate when the top-1 validation accuracy does not improve for 5 epochs and stop training when it plateaus for 10 epochs or when the number of epochs reaches 90. We use a batch size of 256 for all methods. We use cross-validation and find that the optimal initial learning rate is 0.1 for all compared methods. We employ random horizontal flip, random resizing (256-480) with preserved aspect ratio, random crop (224), and color augmentation for data augmentation during training (Krizhevsky et al., 2017). To address the covariant shifts between hidden layers, we employ input mean normalization (IMN) for GmP and mean batch normalization (MBN) for WN. Table 1 reports the single-center-crop top-1 and top-5 validation accuracy for all compared methods, which shows that GmP+IMN significantly outperforms BN and WN+MBN in terms of both top-1 and top-5 validation accuracy. This demonstrates that our method is useful for improving large-scale residual network training.

**Ablation study.** We perform ablation study to provide further insights into how the batch size and intermediate normalization layer affect the convergence speed and generalization performance of different parameterizations. To maintain a manageable computational cost, we conduct these experiments with a medium-sized convolutional neural network VGG-6 (Simonyan & Zisserman, 2014) on ImageNet32 (Chrabaszcz et al., 2017), which contains all 1.3M images and 1,000 categories from ImageNet (ILSVRC 2012) (Deng et al., 2009), but with the images resized to $32 \times 32$. We follow exactly the same experimental setup for optimization and data augmentation as in Chrabaszcz et al. (2017). We use the same optimizer and learning rate scheduler as in the previous experiment.

Table 2: Top-1 and top-5 validation accuracy (%) for VGG-6 trained on ImageNet32.

| Metric | Top-1 validation accuracy | | | Top-5 validation accuracy | | |
|---|---|---|---|---|---|---|
| Batch size | 256 | 512 | 1024 | 256 | 512 | 1024 |
| SP | $38.31 \pm 0.13$ | $36.99 \pm 0.11$ | $35.02 \pm 0.03$ | $62.48 \pm 0.14$ | $60.71 \pm 0.18$ | $58.14 \pm 0.39$ |
| WN | $39.13 \pm 0.10$ | $37.92 \pm 0.12$ | $36.17 \pm 0.03$ | $63.28 \pm 0.02$ | $61.93 \pm 0.09$ | $60.16 \pm 0.18$ |
| WN+MBN | $42.22 \pm 0.01$ | $40.96 \pm 0.02$ | $39.33 \pm 0.07$ | $66.04 \pm 0.07$ | $65.08 \pm 0.03$ | $63.32 \pm 0.08$ |
| BN | $42.79 \pm 0.03$ | $41.90 \pm 0.19$ | $41.39 \pm 0.02$ | $67.17 \pm 0.08$ | $66.50 \pm 0.25$ | $65.89 \pm 0.06$ |
| **GmP** | $40.76 \pm 0.09$ | $41.65 \pm 0.09$ | $41.29 \pm 0.08$ | $65.08 \pm 0.08$ | $65.76 \pm 0.05$ | $65.49 \pm 0.06$ |
| **GmP+IMN** | $\mathbf{43.14 \pm 0.05}$ | $\mathbf{43.62 \pm 0.08}$ | $\mathbf{42.70 \pm 0.15}$ | $\mathbf{67.36 \pm 0.05}$ | $\mathbf{67.76 \pm 0.09}$ | $\mathbf{66.98 \pm 0.18}$ |

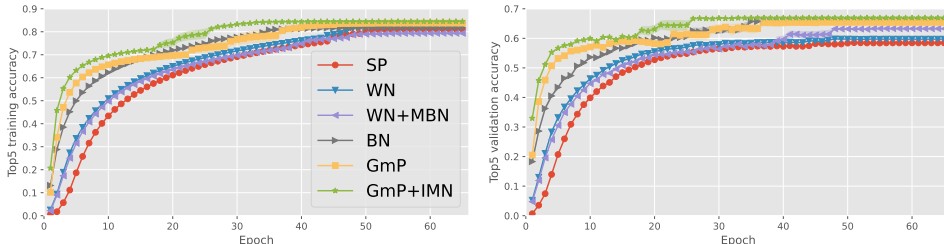

Figure 4: Convergence rate comparison: mean top-5 training and validation accuracy with standard error as a function of training epoch for VGG-6 network trained on the ImageNet32 dataset with a batch size of 1024. Left: top-5 training accuracy. Right: top-5 validation accuracy.

Table 3: Test RMSE for MLP-1 trained on six UCI benchmarks.

| Benchmark | Boston | Concrete | Energy | Power | Wine | Yacht |
|---|---|---|---|---|---|---|
| SP | $3.370 \pm 0.145$ | $5.472 \pm 0.144$ | $0.898 \pm 0.274$ | $4.065 \pm 0.029$ | $0.623 \pm 0.008$ | $0.639 \pm 0.063$ |
| WN | $3.459 \pm 0.156$ | $5.952 \pm 0.148$ | $2.093 \pm 0.789$ | $4.073 \pm 0.026$ | $0.632 \pm 0.008$ | $0.624 \pm 0.076$ |
| BN | $3.469 \pm 0.153$ | $5.695 \pm 0.160$ | $1.648 \pm 0.302$ | $4.164 \pm 0.026$ | $0.622 \pm 0.011$ | $0.777 \pm 0.055$ |
| **GmP** | $\mathbf{3.057 \pm 0.144}$ | $\mathbf{5.153 \pm 0.098}$ | $\mathbf{0.474 \pm 0.013}$ | $\mathbf{4.022 \pm 0.025}$ | $\mathbf{0.613 \pm 0.006}$ | $\mathbf{0.584 \pm 0.046}$ |

We use cross-validation and find that the optimal initial learning rate is $0.1$ for GmP and $0.01$ for all the other methods. Table 2 shows that GmP+IMN consistently achieves the best top-1 and top-5 validation accuracy for all batch sizes considered. Furthermore, the improvement of GmP+IMN over other methods gets larger as the batch size increases, highlighting the robustness and scalability of GmP with large batch sizes. In addition to achieving the best performance, Figure 4 shows that GmP+IMN (the green curve) also converges significantly faster than other compared methods: its top-5 validation accuracy converges within 25 epochs, which is 10 epochs earlier than the second best method BN. The ablation study GmP vs GmP+IMN shows that IMN significantly improves the performance of GmP, which is expected since it addresses the problem of covariant shifts between hidden layers. Notably, Wide ResNet (WRN 28-2) (Zagoruyko & Komodakis, 2016) trained with BN and batch size 500 only achieved $43.08\%$ top-1 validation accuracy as reported in Chrabaszcz et al. (2017), underperforming VGG-6 trained with GmP+IMN ($43.62\%$ as shown in Table 2). This reveals the significance of better parameterizations: *even a small non-residual network like VGG-6 with GmP+IMN can outperform large, wide residual networks like WRN 28-2.*

**UCI Regression with MLP.** To obtain a complete picture of GmP's empirical performance, we also evaluate GmP on six UCI regression datasets (Dua & Graff, 2017), since the same method may exhibit different behaviors on regression tasks and classification tasks. We train an MLP with one hidden layer and 100 hidden units for 10 different random 80/20 train/test splits. We use the Adam optimizer (Kingma & Ba, 2014). We use cross-validation and find that the optimal learning rate is $0.1$ for GmP and $0.01$ for all the other methods. Table 3 shows that GmP consistently achieves the best test RMSE on all benchmarks, significantly outperforming other methods in most cases.

## 6    CONCLUSION

We have presented a novel method, characteristic activation value analysis, for understanding various normalization techniques and their roles in ReLU network feature learning. This method exploits special activation values to characterize ReLU units. The preimage of such characteristic activation values, referred to as the characteristic activation set, is used to identify ReLU units uniquely. To advance the understanding of neural network normalization techniques, we have performed a perturbation analysis for the characteristic activation sets and discovered the instabilities of existing approaches. Motivated by the newly gained insights, we have proposed a new parameterization in the hyperspherical coordinate system called geometric parameterization. We have demonstrated its advantages for single-hidden-layer ReLU networks and combined it with input mean normalization to handle covariance shifts in multiple-hidden-layer ReLU networks. We have performed theoretical analysis and empirical evaluations to validate its usefulness for improving feature learning. We have shown that it consistently and significantly improves training stability, convergence speed, and generalization performance for models of different sizes on a variety of real-world tasks and datasets, including a performance boost to the gold-standard network ResNet-50 on ImageNet (ILSVRC 2012). Limitations and potential future work directions are discussed in Appendix D.

## ETHICS STATEMENT

This paper studies the theory that underpins deep learning, as such it takes a step towards improving the reliability and robustness of deep learning techniques. We believe that the ethical implications of this work are minimal: this research involves no human subjects, no sensitive data where privacy is a concern, no domains where discrimination/bias/fairness is concerning, and is unlikely to have a noticeable social impact. Optimistically, our hope is that this work can produce and inspire better deep learning training algorithms. However, as with most research in machine learning, new modeling and inference techniques could be used by bad actors to cause harm more effectively, but we do not see how this work is more concerning than any other work in this regard.

## REPRODUCIBILITY STATEMENT

The propose method is evaluated on standard, publicly available machine learning benchmarks. The detailed setup for all experiments can be found in Appendix C. The code is submitted as supplementary materials.

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

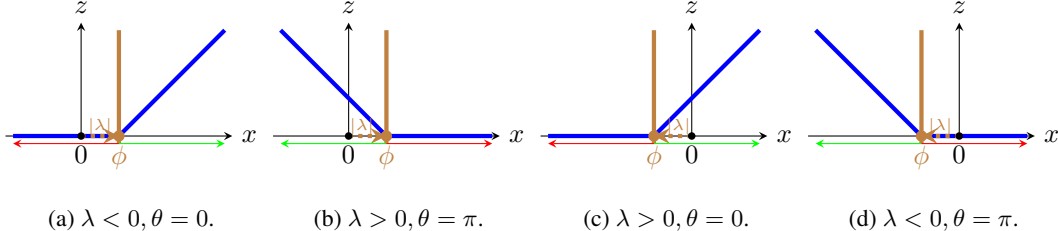

(a) $\lambda < 0, \theta = 0$.     (b) $\lambda > 0, \theta = \pi$.     (c) $\lambda > 0, \theta = 0$.     (d) $\lambda < 0, \theta = \pi$.

Figure 5: Visualization of characteristic activation boundaries $\mathcal{B}$ (brown solid lines) and spatial locations $\phi = -\lambda u(\theta)$ of a single ReLU neuron $z = \mathrm{ReLU}(u(\theta)x + \lambda)$ (blue solid lines) for inputs $x \in \mathbb{R}$. Green arrows denote active regions and red arrows denote inactive regions.

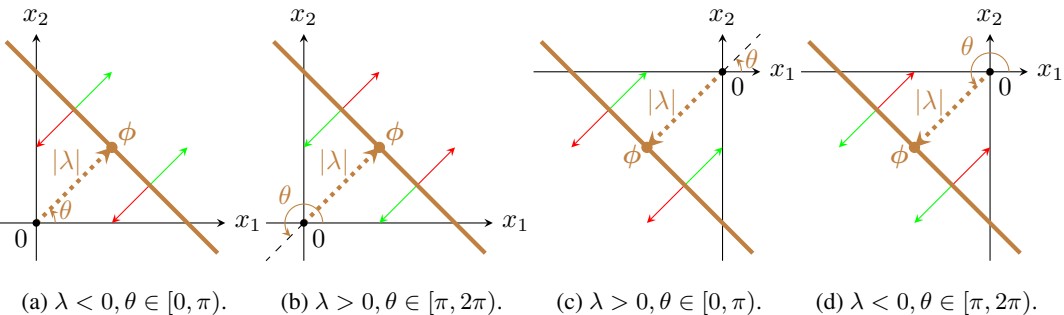

(a) $\lambda < 0, \theta \in [0, \pi)$.    (b) $\lambda > 0, \theta \in [\pi, 2\pi)$.    (c) $\lambda > 0, \theta \in [0, \pi)$.    (d) $\lambda < 0, \theta \in [\pi, 2\pi)$.

Figure 6: Visualization of characteristic activation boundaries $\mathcal{B}$ (brown solid lines) and spatial locations $\boldsymbol{\phi} = -\lambda \, \mathbf{u}(\theta)$ of a single ReLU neuron $z = \mathrm{ReLU}(\mathbf{u}(\theta)^{\mathrm{T}} \mathbf{x} + \lambda) = \mathrm{ReLU}(\cos(\theta)x_1 + \sin(\theta)x_2 + \lambda)$ for inputs $\mathbf{x} \in \mathbb{R}^2$. Green arrows denote active regions and red arrows denote inactive regions.

## A  VISUALIZATIONS OF THE CHARACTERISTIC ACTIVATION BOUNDARIES

Figures 5 and 6 respectively visualize the characteristic activation boundaries in the input spaces $\mathbb{R}$ and $\mathbb{R}^2$ under different conditions of the radius and angle parameters.

## B  PROOF OF THEOREM 3.3

*Proof.* The main idea of this proof comes from differential geometry that the change in the angular direction is the norm of the perturbation with respect to the metric tensor $\mathbf{M}$ for the hyperspherical coordinate system: $\langle \mathbf{u}(\boldsymbol{\theta}), \mathbf{u}(\boldsymbol{\theta} + \boldsymbol{\varepsilon}) \rangle = \|\boldsymbol{\varepsilon}\|_{\mathbf{M}}$. Therefore, we need to work out a formula for calculating $\mathbf{M}$.

Let us start with an input $\mathbf{x} = [x_1, \cdots, x_n]^{\mathrm{T}} \in \mathbb{R}^n$ ($n \geq 2$) in the Cartesian coordinate system, where the metric tensor is the Kronecker delta $m'_{ij} = \delta_{ij}$. For the geometric parameterization of the unit hypersphere $S^{n-1}$, we have

$$
\begin{aligned}
u_1(\boldsymbol{\theta}) &= \cos(\theta_1), \\
u_2(\boldsymbol{\theta}) &= \sin(\theta_1)\cos(\theta_2), \\
u_3(\boldsymbol{\theta}) &= \sin(\theta_1)\sin(\theta_2)\cos(\theta_3), \\
&\vdots \\
u_{n-2}(\boldsymbol{\theta}) &= \sin(\theta_1)\sin(\theta_2)\sin(\theta_3)\cdots\sin(\theta_{n-2})\cos(\theta_{n-1}), \\
u_{n-1}(\boldsymbol{\theta}) &= \sin(\theta_1)\sin(\theta_2)\sin(\theta_3)\cdots\sin(\theta_{n-2})\sin(\theta_{n-1}).
\end{aligned}
\tag{17}
$$

The metric tensor $\mathbf{M}$ for the geometric parameterization of $S^{n-1}$ is the pullback of the Euclidean metric in $\mathbb{R}^n$:

$$m_{ab} = \sum_{i=1}^{n-1} \sum_{j=1}^{n-1} m'_{ij} \frac{\partial u_i}{\partial \theta_a} \frac{\partial u_j}{\partial \theta_b} = \sum_{i=1}^{n-1} \frac{\partial u_i}{\partial \theta_a} \frac{\partial u_i}{\partial \theta_b}. \tag{18}$$

- $a \neq b$: we have $m_{ab} = 0$, since it is a sum of terms that are either zero or with alternating signs which cancel out. Hence, $\mathbf{M}$ is a diagnoal matrix.

- $a = b$: Using the fact that $\sin^2(\theta_j) + \cos^2(\theta_j) = 1, \forall j$, we have $m_{11} = 1$ and

$$m_{aa} = \sum_{i=1}^{a-1} \sin^2(\theta_i), \quad 2 \leq a \leq n-1. \tag{19}$$

Therefore, the metric tensor for the hyperspherical coordinate is a diagonal matrix

$$\mathbf{M} = \begin{bmatrix} 1 & 0 & 0 & \cdots & 0 & 0 \\ 0 & \sin^2(\theta_1) & 0 & \cdots & 0 & 0 \\ 0 & 0 & \sin^2(\theta_1) + \sin^2(\theta_2) & \cdots & 0 & 0 \\ \vdots & \vdots & \vdots & \ddots & \vdots & \vdots \\ 0 & 0 & 0 & \cdots & \sum_{i=1}^{n-2} \sin^2(\theta_i) & 0 \\ 0 & 0 & 0 & \cdots & 0 & \sum_{i=1}^{n-1} \sin^2(\theta_i) \end{bmatrix}. \tag{20}$$

Finally, the change in the angular direction of a unit vector $\mathbf{u}(\boldsymbol{\theta})$ under a perturbation $\boldsymbol{\varepsilon}$ to $\boldsymbol{\theta}$ is given by the norm of $\boldsymbol{\varepsilon}$ with respect to the tensor matrix $\mathbf{M}$:

$$\langle \mathbf{u}(\boldsymbol{\theta}), \mathbf{u}(\boldsymbol{\theta} + \boldsymbol{\varepsilon}) \rangle = \|\boldsymbol{\varepsilon}\|_{\mathbf{M}} = \sqrt{\boldsymbol{\varepsilon}^{\mathbf{T}} \mathbf{M} \boldsymbol{\varepsilon}} = \sqrt{\varepsilon_1^2 + \sum_{i=2}^{n-1} \left( \prod_{j=1}^{i-1} \sin^2(\theta_j) \right) \varepsilon_i^2}. \tag{21}$$

Moreover, since $0 \leq m_{ii} = \prod_{j=1}^{i-1} \sin^2(\theta_j) \leq 1$ for all $i$, we have that

$$\langle \mathbf{u}(\boldsymbol{\theta}), \mathbf{u}(\boldsymbol{\theta} + \boldsymbol{\varepsilon}) \rangle \leq \sqrt{\varepsilon_1^2 + \sum_{i=2}^{n-1} \varepsilon_i^2} = \|\boldsymbol{\varepsilon}\|_2. \tag{22}$$

This completes the proof. $\square$

## C  DETAILED EXPERIMENTAL SETUPS

### C.1  IMAGENET CLASSIFICATION WITH RESNET-50

We train a ResNet-50 (He et al., 2016) on the ImageNet (ILSVRC 2012) dataset (Deng et al., 2009), which consists of 1,281,167 training images and 50,000 validation images that contain objects from 1,000 categories. The size of the images ranges from $75 \times 56$ to $4288 \times 2848$. We follow exactly the same experimental setup for optimization and data augmentation as in He et al. (2016). We use the SGD optimizer with momentum 0.9, which turns out to be better than Adam for image classification tasks (He et al., 2016). We reduce the learning rate when the top-1 validation accuracy does not improve for 5 epochs and stop training when it plateaus for 10 epochs or when the number of epochs reaches 90. We use a batch size of 256 for all methods. We use cross-validation to select the learning rate for each compared method from the set $\{0.001, 0.003, 0.01, 0.03, 0.1, 0.3\}$. We find that the optimal initial learning rate is $0.1$ for all compared methods. We employ random horizontal flip, random resizing (256-480) with preserved aspect ratio, random crop (224), and color augmentation for data augmentation during training (Krizhevsky et al., 2017). To address the covariant shifts between hidden layers, we employ input mean normalization (IMN) for GmP and mean batch normalization (MBN) for WN. We report single-center-crop top-1 and top-5 validation accuracy.

## C.2 Ablation Study: ImageNet32 Classification with VGG-6

To maintain a manageable computational cost for the ablation study, we train a VGG-6 (Simonyan & Zisserman, 2014) on ImageNet32 (Chrabaszcz et al., 2017), which contains all 1.3M images and 1,000 categories from ImageNet (ILSVRC 2012) (Deng et al., 2009), but with the images resized to $32 \times 32$. We follow exactly the same experimental setup for optimization and data augmentation as in Chrabaszcz et al. (2017). We use the SGD optimizer with momentum 0.9, which turns out to be better than Adam for image classification tasks (He et al., 2016). We reduce the learning rate when the top-1 validation accuracy does not improve for 5 epochs and stop training when it plateaus for 10 epochs or when the number of epochs reaches 90. We train the model using three common batch sizes $\{256, 512, 1024\}$ for all methods. We use cross-validation to select the learning rate for each compared method from the set $\{0.001, 0.003, 0.01, 0.03, 0.1, 0.3\}$. We find that the optimal initial learning rate is 0.1 for GmP and 0.01 all the other methods. We employ random horizontal flips for data augmentation during training. We conduct an ablation study to explore the effects of input mean normalization (IMN) for GmP and mean batch normalization (MBN) for WN in deep networks. We report top-1 and top-5 validation accuracy.

## C.3 UCI Regression with MLP

We train an MLP with one hidden layer and 100 hidden units for 10 different random 80/20 train/test splits. We use the Adam optimizer (Kingma & Ba, 2014) with full-batch training. We use cross-validation to select the learning rate for each compared method from the set $\{0.001, 0.003, 0.01, 0.03, 0.1, 0.3\}$. We find that the optimal initial learning rate is 0.1 for GmP and 0.01 for all the other compared methods. We report test root mean squared error (RMSE).

# D Limitations and Future Work

This work analyzed the ReLU activation function due to its wider adoption. However, the general characteristic value analysis technique can be extended to other activation functions. Also, we only performed the characteristic activation analysis for single-hidden-layer ReLU networks and proposed a practical workaround to address the problem of covariant shift between hidden layers by using input mean normalization. For future work, this analysis needs to be generalized to examine feature learning in multiple-hidden-layer networks to understand the theoretical behavior of deep networks. One potential difficulty with multiple-hidden-layer networks is that the characteristic activation boundary becomes a piecewise linear partition of the input space, which is less straightforward to analyze. A possible solution would be to consider how the assignment of each data point to the partition evolves during training, similar to how we track the characteristic activation sets.

