# OpenReview forum: "Neural Characteristic Activation Value Analysis for Improved ReLU Network Feature Learning"
_ICLR.cc/2024/Conference — Submitted to ICLR 2024_

### Official Review · Reviewer_n38z · 2023-10-15

**Soundness:** 3 good
**Presentation:** 3 good
**Contribution:** 3 good
**Rating:** 6
**Confidence:** 2

**Summary:**

This paper proposed a novel approach to understanding and improving ReLU-based neural networks. The authors delve into the characteristic activation values of individual ReLU units within neural networks and establish a connection between these values and the learned features. They propose a geometric parameterization for ReLU networks based on hyperspherical coordinates, which separates radial and angular parameters. This new parameterization is demonstrated to enhance optimization stability, convergence speed, and generalization performance.

**Strengths:**

- This paper is written in a clear and easily comprehensible manner, making it easy for readers to follow.
- The paper presents a unique and innovative approach to understanding ReLU networks by exploring the characteristic activation values. This fresh perspective sheds light on the inner workings of these networks, offering insights that were previously unexplored.

**Weaknesses:**

see questions.

**Questions:**

- Can the analysis apply to the existing advanced batch normalization improvements like IEBN [1], SwitchNorm [2], layer norm [3]. These missing works should be considered and added to the related works or analysis.

- I am not very familiar with the topics covered in this article, I will consider these clarifications along with feedback from other reviewers in deciding whether to raise my score.

[1] Instance Enhancement Batch Normalization: An Adaptive Regulator of Batch Noise, AAAI

[2] Differentiable Learning-to-Normalize via Switchable Normalization, ICLR

[3] Layer normalization, IJCAI

---

> ### Author Response · Authors · 2023-11-16
> **Response to Reviewer n38z**
>
> Thanks for your comments on our paper! We will respond to your comments point by point below:
>
> > Can the analysis apply to the existing advanced batch normalization improvements like IEBN [1], SwitchNorm [2], layer norm [3]? These missing works should be considered and added to the related works or analysis.
>
> Thanks for pointing out these related works. Please note that LayerNorm is already discussed in the related work section (Section 2) under “Other normalization methods”. We have added [1,2] to the related work section in the updated manuscript.
>
> Regarding the applicability of our analysis, the proposed characteristic activation value analysis is designed to analyze neural network parameterizations. We showed in Section 2 that BatchNorm can be viewed as a kind of weight normalization, so our analysis will produce the same result for BatchNorm as that for weight normalization. Furthermore, since BatchNorm, IEBN, SwitchNorm, LayerNorm, and almost all existing normalization techniques are different variants of the same normalization technique which operate in the weight-space parameterization, they will have the same instability property as standard parameterization and weight normalization, as shown in Equations 12 and 14. We have clarified this in the updated paper manuscript.
>
> We hope that this has sufficiently addressed all your concerns. Please let us know if you have any further questions or comments.

---

> > ### Comment · Reviewer_n38z · 2023-11-23
> > **Thank you for your response.**
> >
> > Thank you for your detailed response; my concern has been addressed. I am inclined to keep my positive scores. Thank you for your efforts.

---

> > > ### Author Response · Authors · 2023-11-23
> > > **Thank you for acknowledging our response**
> > >
> > > Thank you for acknowledging our response! We are glad that your concern has been addressed.

---

### Official Review · Reviewer_cECr · 2023-11-04

**Soundness:** 2 fair
**Presentation:** 3 good
**Contribution:** 2 fair
**Rating:** 3
**Confidence:** 3

**Summary:**

This paper proposes a geometric parameterization method for ReLU networks to improve their performance. Some experimental results show the performance of the proposed method.

**Strengths:**

1. This paper proposes a geometric parameterization method for ReLU networks to improve their performance.
2. Some experimental results show the performance of the proposed method.

**Weaknesses:**

Although the paper is theoretically and experimental sound, there are still some questions need to be discussed in this paper:
1.	The main contributions of this paper are to propose one geometric parameterization for ReLU networks and input mean normalization. But the input mean normalization proposed in this paper is very similar to the mean-only batch normalization (Salimans & Kingma, 2016). What’s the advantage of the former against the latter?
2.	The experimental results are not convincing. The authors should compare the performance of the proposed algorithm on more models and datasets.
3.	Both the English language and equations in this paper need to be improved.

**Questions:**

Although the paper is theoretically and experimental sound, there are still some questions need to be discussed in this paper:
1.	The main contributions of this paper are to propose one geometric parameterization for ReLU networks and input mean normalization. But the input mean normalization proposed in this paper is very similar to the mean-only batch normalization (Salimans & Kingma, 2016). What’s the advantage of the former against the latter?
2.	The experimental results are not convincing. The authors should compare the performance of the proposed algorithm on more models and datasets.
3.	Both the English language and equations in this paper need to be improved.

---

> ### Author Response · Authors · 2023-11-16
> **Response to Reviewer cECr**
>
> Thanks for your comments on our paper! We will respond to your comments point by point below:
>
> > The main contributions of this paper are to propose one geometric parameterization for ReLU networks and input mean normalization. But the input mean normalization proposed in this paper is very similar to the mean-only batch normalization (Salimans & Kingma, 2016). What’s the advantage of the former against the latter?
>
> Regarding our contributions, please note that geometric parameterization and input mean normalization are only part of the contributions in our paper, which are natural products from our main contribution - the characteristic activation value analysis for neural network parameterizations. We used the proposed novel analysis to identify the instability of common neural network parameterizations such as standard parameterization and weight normalization. Motivated by this, we proposed geometric parameterization which has theoretical stability guarantees as shown by our analysis in Theorem 3.3. We verified the theoretical result with three illustrative experiments in Section 3.5.
>
> Regarding input mean normaliztion vs mean batch normalization, as discussed at the beginning of Section 4.2, the input mean normalization is a technique to address the covariate shift problem in multi-layer networks with geometric parameterization. We subtract the mean from the input to each layer because our proposed analysis assumes that input distribution is centered around the origin. The mean-only batch normalization (Salimans & Kingma, 2016) would not address this problem because it is applied to the pre-activation values (i.e., in this case the zero mean distribution will be destroyed after going through the ReLU activation, so the input distribution to the next layer will not be zero mean anymore). We have clarified this in the updated manuscript.
>
> > The experimental results are not convincing. The authors should compare the performance of the proposed algorithm on more models and datasets.
>
> We have conducted extensive experiments with
> - three representative models: MLP, VGG, and ResNet;
> - several representative regression and classification benchmarks: ImageNet (ILSVRC 2012) classification dataset, ImageNet-32 classification dataset, 6 UCI regression datasets, 2D Banana classification dataset, and 1D Levy regression dataset.
>
> We believe that these experiments have verified the usefulness of the proposed algorithm with different model architectures and on benchmarks from different domains.
>
> > Both the English language and equations in this paper need to be improved.
>
> We’d be grateful if the reviewer could point out the texts or equations that are confusing, and we will try our best to clarify them and improve our presentation. That said, we noticed that Reviewer n38z actually appreciated our clear writing in an easily comprehensible manner.
>
> We hope that this has sufficiently addressed all your concerns. Please let us know if you have any further questions or comments.

---

> ### Author Response · Authors · 2023-11-23
>
> Thanks again for your effort during the reviewing process! We believe that we have addressed your stated concerns in our response and would like to ask if the reviewer thinks this is the case as well. If our response is affirmative, we would like to ask the reviewer if they are happy to increase their scores.

---

### Official Review · Reviewer_BSGv · 2023-11-05

**Soundness:** 3 good
**Presentation:** 3 good
**Contribution:** 3 good
**Rating:** 6
**Confidence:** 3

**Summary:**

The paper proposes geometric parameterization (GmP) for ReLU networks.

**Strengths:**

1. This paper introduced the characteristic activation sets of individual neurons and a geometric connection between such sets and learned features in ReLU networks.
2. This paper then introduced geometric parameterization (GmP) based on radial-angular decomposition in the hyperspherical coordinate system. It also proves that the change in the angular direction under perturbation $\varepsilon$ is bounded by the magnitude of perturbation $\varepsilon$. This property is not held for standard parameterization and weight normalization.
3. The authors provide some experimental results to show the advantage of the proposed GmP for ReLU networks.

**Weaknesses:**

1. The Gmp for the ReLU network with IMN in Eq.16 also seems applicable to the weight normalization (WN), i.e. change $u(\theta)$ to $\frac{w}{\||w\||_2}$. I am wondering what the result will look like. The authors should talk more about why they prefer optimization in the angular space instead of the weight normalization space since they are equivalent as shown in Eq. 9 ( $u(\theta):=\frac{w}{\||w\||_2}$).
2. The theoretical proof only shows that the change in the angular direction under perturbation $\varepsilon$ is bounded by the magnitude of perturbation $\varepsilon$. Using $\Delta \phi$ as evidence for generalization is not theoretically justified. I think loss function values should be added as supporting evidence too.
3. I found some of the experimental settings unsatisfactory. For example, the training settings of ResNet-50 are very different compared to standard settings. I expect to see a comparison of different methods with standard learning rate decay, i.e. first 30 epochs: 0.1, 30-60 epochs: 0.01, 60-90 epochs: 0.001. Under this setting, BN should achieve around 76.1% top-1 accuracy. Or, the authors should provide a convincing explanation why not use the standard training setting.

**Questions:**

1. In Eq.16, only input mean normalization (IMN) is used, why not further normalize the input features with its variance?

---

> ### Author Response · Authors · 2023-11-16
> **Response to Reviewer BSGv**
>
> Thanks for your comments on our paper! We will respond to your comments point by point below:
>
> > The authors should talk more about why they prefer optimization in the angular space instead of the weight normalization space since they are equivalent as shown in Eq. 9
>
> Section 3.4 talks about why optimization is preferred in the angular-space rather than the weight-space, which is the main theoretical contribution of this paper. In summary, Theorem 3.1 shows that the change of angular direction under small perturbation is bounded by the magnitude of perturbation, which is robust against noise during stochastic gradient optimization. In contrast, although Eq. 9 shows an equivalency between weight normalization and geometric parameterization, Eq. 14 shows that they have entirely different stability properties, i.e., weight normalization is as unstable as standard parameterization under small noise perturbation during stochastic gradient optimization. Furthermore, this further implies that any weight-space parameterization will suffer from this issue because they will all lead to Eq. 14. We have clarified this in the updated manuscript.
>
> > The theoretical proof only shows that the change in the angular direction under perturbation is bounded by the magnitude of perturbation. Using $\Delta\phi$ as evidence for generalization is not theoretically justified. I think loss function values should be added as supporting evidence too.
>
> We did not use $\Delta\phi$ as evidence for generalization. Instead, we used it in our illustrative experiments in Section 3.5 to verify that our theoretical results about the stability of different parameterizations from Section 3.4 hold in practice. The evidence we used for generalization is the final test performance (RMSE and accuracy), i.e., Figure 2(d)-(g), Figure 3(j)-(m), Figure 4, and Tables 1, 2 and 3. The loss function values are consistent with these test performance, which we didn’t report in the paper due to page limits.
>
> > I found some of the experimental settings unsatisfactory. For example, the training settings of ResNet-50 are very different compared to standard settings. I expect to see a comparison of different methods with standard learning rate decay, i.e. first 30 epochs: 0.1, 30-60 epochs: 0.01, 60-90 epochs: 0.001. Under this setting, BN should achieve around 76.1% top-1 accuracy. Or, the authors should provide a convincing explanation why not use the standard training setting.
>
> This is because different parameterizations have different convergence speeds as shown in our ablation study (Figure 4). The fixed learning rate decay scheme mentioned by the reviewer may be useful for BatchNorm, but it can be suboptimal and thus slow down the convergence of geometric parameterization (because it converges faster and will reach the plateau faster at each stage than BatchNorm). Therefore, we used an adaptive learning rate decay scheme, Reduce LR On Plateau, for geometric parameterization. Then, we had to use Reduce LR On Plateau for BatchNorm and weight normalization as well for a fair comparison.
>
> > In Eq.16, only input mean normalization (IMN) is used, why not further normalize the input features with its variance?
>
> IMN is used to address the covariate shift problem in multi-layer networks with geometric parameterization, which ensure that the input distributions to the intermediate layers of multi-layer networks are centered around the origin, because this is an assumption of our proposed analysis. There is no need to further normalize the input features with its variance because our analysis does not assume that the input distribution has unit variance. That said, it would be harmless to normalize the variance in terms of the final test performance, but calculating the variance for the input to each layer in each forward pass would increase the computational cost and slow down training, and therefore we chose not to do that. We have clarified this in the updated manuscript.
>
> We hope that this has sufficiently addressed all your concerns. Please let us know if you have any further questions or comments.

---

> ### Author Response · Authors · 2023-11-23
>
> Thanks again for your effort during the reviewing process! We believe that we have addressed your stated concerns in our response and would like to ask if the reviewer thinks this is the case as well. If our response is affirmative, we would like to ask the reviewer if they are happy to champion our paper.

---

### Author Response · Authors · 2023-11-16
**General Response**

We thank all reviewers for their valuable comments and feedback on our paper. We have addressed each reviewer’s concerns separately below their respective review. Please let us know if you have further questions or comments. We also have updated our manuscript, where changes are highlighted in blue.

---

### Meta-Review · Area_Chair_oPAM · 2023-12-21

**Metareview:**

After a thorough evaluation of the reviews provided by Reviewers BSGv, cECr, and n38z, as well as the authors' rebuttals and subsequent discussions, the decision is to reject Paper 4349.

Reviewer BSGv noted strengths in the conceptual framing of the paper and the introduction of geometric parameterization (GmP) for ReLU networks. However, concerns were raised about the experimental settings and the need for further clarification on the preference for optimization in the angular space versus weight normalization space.

Reviewer cECr expressed doubts regarding the novelty and significance of the contributions, particularly the input mean normalization technique, which appears similar to existing approaches such as mean-only batch normalization. The reviewer also found the experimental results unconvincing and suggested that more models and datasets should be tested to validate the proposed method. Furthermore, the reviewer called for improvements in the clarity of the English language and equations.

While Reviewer n38z was generally positive about the clarity of the presentation and the fresh perspective on ReLU networks, there were questions about the applicability of the analysis to other advanced normalization techniques, which the authors addressed in their response.

Despite the authors' efforts to clarify their contributions and respond to the reviews, significant concerns remain. Reviewer cECr's request for more specific experiments and clarity improvements has not been adequately addressed, and there is no evidence that the reviewer's stance has changed post-rebuttal.

Given the mixed reviews and the fact that the authors have not convincingly addressed the main concerns regarding the experimental validation and differentiation from existing techniques, the paper does not meet the acceptance criteria for publication at this time. It is essential for the authors to engage more deeply with the critiques and provide substantial additional evidence and clarification to strengthen the paper's contributions and experimental validation.

**Justification For Why Not Higher Score:**

The decision to reject is not taken lightly and is based on the balance of all reviews. While there is recognition of some of the paper's strengths, particularly in theoretical conceptualization and attempts at innovation, the weaknesses identified by the reviewers, especially in terms of experimental validation and differentiation from existing work, are significant.

The score is not higher because the authors have not fully addressed the critical concerns raised by the reviewers. More extensive experimentation, clearer exposition, and a more convincing argument for the advantages of the proposed method over existing normalization techniques are needed.

**Justification For Why Not Lower Score:**

N/A

---

### Decision · Program_Chairs · 2024-01-16

Reject